# Chiral Mechanical Effect of the Tightly Focused Chiral Vector Vortex Fields Interacting with Particles

**DOI:** 10.3390/nano13152251

**Published:** 2023-08-04

**Authors:** Qiang Zhang, Zhirong Liu, Ziqiang Cheng

**Affiliations:** Department of Applied Physics, East China Jiaotong University, Nanchang 330013, China

**Keywords:** chiral mechanical effect, superchiral focused field, particle sorting and manipulation

## Abstract

The coupling of the spin-orbit angular momentum of photons in a focused spatial region can enhance the localized optical field’s chirality. In this paper, a scheme for producing a superchiral optical field in a 4π microscopic system is presented by tightly focusing two counter-propagating spiral wavefronts. We calculate the optical forces and torques exerted on a chiral dipole by the chiral light field and reveal the chiral forces by combining the light field and dipoles. Results indicate that, in addition to the general optical force, particles’ motion would be affected by a chiral force that is directly related to the particle chirality. This chiral mechanical effect experienced by the electromagnetic dipoles excited on a chiral particle could be characterized by the behaviors of chirality density and flux, which are, respectively, associated with the reactive and dissipative components of the chiral forces. This work facilitates the advancement of optical separation and manipulation techniques for chiral particles.

## 1. Introduction

Chirality is defined as a structure that is not superimposable with its mirror image [1], which manifests throughout nature. In addition to molecular chirality, there is also the chirality of light, best exemplified by circularly polarized light (CPL) [2]. Chirality of light results from the electric and magnetic vector fields that swirl as the wave propagates [3]. Light can carry orbital angular momentum (OAM), creating a helical geometry on the wavefront. Thus, the chirality that OAM light carries could be elucidated by some chiroptical parameters, such as the spin-density fluxes ($L_{e}$ and $L_{m}$), optical helicity fluxes ($\Phi_{\chi }$ and $\Phi_{\hbar }$), and orbital and spin angular momentum ($L_{z}$ and $S_{z}$). Analogously, the chirality of an object can be quantified as the chirality parameter (χ) and chiral polarization tensor ($\alpha_{c})$ [4,5].

In recent years, intensive research has been developed on the construction of chiral optical fields, such as controllable chiral light fields with OAM [6,7,8], superchiral optical fields that have enhanced optical chirality [9,10,11,12], spatially structured chiral optical patterns with localized optical chirality [13,14], and synthetic chiral optical fields that serve specific demands [15]. Tang et al. introduced the concept of optical chirality as a measure of the localized chirality density of the electromagnetic field and expounded on the “superchirality” as a means of enhancing the enantioselectivity of some chiroptical techniques [16,17]. In this context, extensive research on the measurement and identification of optical chirality has pushed novel notions, such as the chirality density and flow, to the forefront of optics from both theoretical and practical standpoints [18,19,20,21]. The spin and orbital angular momentum of photons endow the vortex light field with optical chirality [22], which begets a spate of intriguing phenomena in the interaction between the OAM light and chiral substances, such as spin–orbit coupling and vortex circular dichroism (VCD) [23,24,25,26]. Theoretically, the infinite-dimensional OAM states provide a broader method of measuring the optical chirality of structures with a monochromatic laser, which essentially promotes the chiral response in the interaction between chiral light and matter [15,27,28,29]. The spiral phase wavefront (photon OAM) contained in the vortex light could interact with the chiral microstructure to generate gigantic vortical differential scattering (VDS) by matching the dimension of the vortex beam and the chiral structure [30,31,32,33]. This demonstrates the possibility of utilizing OAM light to detect the optical chirality of structures with varying geometric properties [34,35]. The OAM states induced by chirality would promote the superchiral response of photon spin states to excited molecules and realize the exact capture of nanoparticles with complex trajectories [36,37,38,39,40,41,42]. Up to now, research on the chiral mechanical effect has only considered intrinsic chirality while ignoring the exterior chiral effects in achiral systems. Hence, we expect to provide an obvious appreciation for the interaction between light and chiral substances and further open up new avenues for optical manipulation.

In this work, we present a 4π optical microscopic system with two high numerical aperture (NA) objectives developed to improve the focusing efficiency by coherent interference of the two counter-propagating cylindrical vector (CV) vortex beams. The OAM states of the photon are introduced to enhance the VCD of the focused field, thus realizing the localized chirality enhancement of the chiral focused field at the focal position where the chiral particles are located. By altering the sign of the incident photon’s OAM state and chirality factors, the difference in the optical force of the chiral dipole when excited by photons of opposing states can be adjusted to enable chiral sorting. Additionally, the optical force and the moment exerted by the chiral focused field on the chiral dipole are examined, and a novel chiral mechanical effect is revealed from the perspective of the chiral field and particle. These findings may facilitate the construction of a super-chiral optical field and the manipulation and separation of multiple chiral particles [43,44,45].

## 2. Theoretical Model

Note that a purely longitudinal electric field can be generated by focusing two counter-propagating spiral wavefronts with opposite instantaneous polarization in a 4π focusing system due to the destructive interference of the radial and azimuthal components. Figure 1 depicts the 4π focusing system consisting of two identical high numerical aperture (NA) lenses for producing a highly symmetrical chiral focused field. The instantaneous electromagnetic vector of two counter-propagating pupil fields with identical handedness passes through the focusing system, where the chirality flows are canceled out because the electric and magnetic fields are orthogonal. Due to its high focusing efficiency, the device could be utilized to manipulate particles and generate sustainable three-dimensional optical trapping.

Here, we introduce a CV vortex beam, whose polarization vector can subsequently be resolved into radial and azimuthal components $s=\cos\varphi_{0}e_{\rho }+\sin\varphi_{0}e_{\varphi }$, where $\varphi_{0}=\frac{\pi }{4}$ denotes the polarization angle and $\left(e_{\rho },e_{\varphi }\right)$ represents the unit vector corresponding to the object plane’s radial and azimuthally polarized directions. On the initial plane, the electric field can be expressed as
(1)$$E_{a}\left(\theta ,\varphi \right)=\sqrt{\cos\theta }\left[E_{i\rho }s+E_{i\varphi }\left(s\times e_{z}\right)\right],$$
with θ indicating the aperture angle of the lens, φ denoting the azimuth of the CV beam, $\left(E_{i\rho },E_{i\varphi }\right)$ representing the radial and azimuthal components of the input field $E_{i}$, and $e_{z}$ being an axial unit vector along the *z* axis. The input field $E_{i}$ at the entrance pupil could be described as the transformation of $E_{a}$:(2)$$E_{i}\left(\theta ,\varphi \right)=l\left(\theta \right)exp\left(im\varphi \right)\left[E_{i\rho }s_{1}+E_{i\varphi }\left(s_{1}\times e_{z}\right)\right],$$
where *m* is the topological charge, l(θ) provides the relative field amplitude function, and $s_{1}$ lies in the plane containing both the ray and the optical axis and is perpendicular to the electric-field propagation direction $e_{z}$, which can be expressed as
(3)$$s_{1}=cos\theta \left(\cos\varphi_{0}e_{\rho }+\sin\varphi_{0}e_{\varphi }\right)+sin\theta e_{z}.$$

Assume that the amplitude function l(θ) has such a form, expressed as $l\left(\theta \right)=J_{s}\left(k_{r}\beta_{0}\frac{\sin\theta }{\sin\alpha }\right)\exp\left[-\beta_{0}{\left(\frac{\sin\theta }{\sin\alpha )}\right)}^{2}\right]$, where the high NA lens interception ratio $\beta_{0}=r$/ω is the ratio of the pupil radius *r* of the lens to the beam waist radius ω of the incident beam.

Given the tight focusing of the incident electric field through a high NA lens with a focal length of *f*, it can be assumed that the origin of the coordinate system is located at the focus of the focusing lens. According to the Richards–Wolf vector diffraction theory, the electric field near the focus can be written as [46]:(4)$$E\left(r\right)=\frac{-ikf}{2\pi }\int_{0}^{\theta_{max}}\int_{0}^{2\pi }E_{i}\left(\theta ,\varphi \right)e^{ik(\hat{s_{1}}\cdot r_{p})}\sin\theta d\theta d\varphi ,$$
where cylindrical coordinates $r=(\rho_{p},\varphi_{p},z_{p})$ are located in the image space for points *p* near the paraxial focus, $\hat{s_{1}}\cdot r_{p}=\sin\rho_{p}\cos\left(\varphi -\varphi_{p}\right)+z_{p}\sin\theta$, k=2nπ/λ is the wave number of the beam in the lens space, the maximum aperture angle $\theta_{max}=\sin^{-1}\left(NA/n\right)$, and *n* is the refractive index of the lens space.

Integrating over the azimuthal direction in Equation (Equation 1) yields three orthogonal electric-field polarization components $\left(E_{\rho },E_{\varphi },E_{z}\right)$ in the cylindrical coordinate system $\left(\rho_{p},\varphi_{p},z_{p}\right)$ of the focused image space. The position vector $\vec{\rho_{p}}$ of the observation point *p* in the space domain could be expressed as $\vec{\rho_{p}}=\rho_{p}\cos\varphi_{p}\hat{x}+\rho_{p}\sin\varphi_{p}\hat{y}+z_{p}\hat{z}$ with $\rho_{p}={\left(x_{p}^{2}+y_{p}^{2}\right)}^{1/2}$, and the azimuth angle $\varphi_{p}$ starting from the point *p* can be written as $\varphi_{p}=\arctan\left(y_{p}/x_{p}\right)$. According to the Debye vector diffraction theory [46,47], the radial, azimuthal, and longitudinal electric-field polarization components after the tight focusing of the CV vortex beam could be described as:(5)$$\begin{array}{c} \left[\begin{array}{c} E_{\rho }\left(\rho_{p},\varphi_{p},z_{p}\right)\\ E_{\varphi }\left(\rho_{p},\varphi_{p},z_{p}\right)\\ E_{z}\left(\rho_{p},\varphi_{p},z_{p}\right) \end{array}\right]=\frac{-ikf}{2\pi }\int_{0}^{\theta_{max}}l(\theta )\sin\theta e^{\left(ikz_{p}\cos\theta \right)}\\ \left[\begin{array}{c} \cos\theta \left[I_{l}^{s}\left(\rho \right)+K_{l}^{s}\left(\rho \right)\right]e_{\rho }\\ \cos\theta \left[iK_{l}^{s}\left(\rho \right)-iI_{l}^{s}\left(\rho \right)\right]e_{\varphi }\\ -\sin\theta \left(2i^{m}e^{i\left(m\right)\varphi_{p}}\right)e_{z} \end{array}\right]d\theta , \end{array}$$
with functions $I_{l}^{s}\left(\rho \right)=i^{m+s}e^{i\left(m+s\right)\varphi_{p}}J_{m+s}\left(k\rho_{p}\sin\theta \right)$ and $K_{l}^{s}\left(\rho \right)=i^{m-s}e^{i\left(m-s\right)\varphi_{p}}$$J_{m-s}$$\left(k\rho_{p}\sin\theta \right)$.

The final expression of the electric field near the focal point in the 4π optical focusing system can be obtained from the left and right objectives as follows:(6)$$E=E_{L}\left(\rho_{p},\varphi_{p},z_{p}\right)+\exp\left(i\Delta \varphi \right)E_{R}\left(\rho_{p},\varphi_{p},-z_{p}\right),$$
where $E_{L}$ and $E_{R}$ are the electric fields focused by the left and right objectives, respectively, and Δφ denotes the phase delay between the two parts. In theory, the π/2 phase delay between the two components can produce a CV vortex focused field with optical chirality.

In order to measure the chirality of the optical field and understand the mechanism underlying its formation, it is necessary to characterize the chiral field by its optical chirality [48]:(7)$$C=\frac{\varepsilon_{0}}{2}E\cdot ▽\times E+\frac{\mu_{0}}{2}H\cdot ▽\times H,$$
and a chirality flow:(8)$$\Phi =\frac{\omega }{2}\left(\varepsilon_{0}\Phi_{E}+\mu_{0}\Phi_{H}\right)=\frac{\varepsilon_{0}}{2}\left(E\times \dot{E}\right)+\frac{\mu_{0}}{2}\left(H\times \dot{H}\right),$$
where $\varepsilon_{0}$ and $\mu_{0}$ denote the vacuum permittivity and permeability, $\Phi_{E}$ and $\Phi_{H}$ are the electric and magnetic ellipticities, and E and H present the electric and magnetic fields, respectively.

Additionally, superchirality is introduced to further reveal the localized chiral enhancement effect of focused vector vortex beams. It is defined as the ratio of the chiral density *g* of the focused vector vortex beams to the amplitude of the CPL, given as [13]:(9)$$g/g_{CL}=Z_{0}\frac{Im(E\cdot H^{*})}{{\left|E\right|}^{2}}>1,$$
with
(10)$$g=\frac{\omega }{2c^{2}}Im\left(E\cdot H^{*}\right),$$
in which $Z_{0}$ is the wave impedance in the vacuum. The numerator in Equation (Equation 9) should satisfy the condition $Im\left(E\cdot H^{*}\right)\leq \left|E\right|\left|H\right|$, resulting in $g/g_{CL}<Z_{0}\left|H\right|/\left|E\right|$. Thus, such a necessary condition $\left|H\right|/\left|E\right|>1/Z_{0}$ that realizes the superchiral field is obtained.

For a minor chiral Rayleigh particle, such as a spherical dipolar particle suspended in the space of the chiral focused field, whose optical properties could be described by the polarizability $a=\alpha /(1-i\alpha k^{3}/6\pi )$ with $\alpha =4\pi r^{3}\left(\varepsilon_{a}-\varepsilon_{0}\right)/\left(\varepsilon_{a}+2\varepsilon_{0}\right)$ [49], where *r* is the radius of the particle, $\varepsilon_{a}$ is the relative permittivity of the chiral Rayleigh particle, and $\varepsilon_{0}$ is the relative permittivity in a background medium.

Within the range of validity of the dipolar approximation, the interaction between particles and (E,H) could be characterized as a coupling of induced electric and magnetic dipole moments [50,51]. Light exerts instantaneous force and momentum on the electric and magnetic dipole [52]. Thus, the vector expression of the optical force acting on the particles can be expressed as [53]:(11)$$\langle F\rangle =\frac{1}{2}\left[\left(\nabla E^{*}\right)p+\left(\nabla B^{*}\right)m-\frac{\beta^{4}}{6\pi }\sqrt{\frac{\mu }{\varepsilon }}\left(p\times m^{*}\right)\right],$$
where the induced electric and magnetic dipole moments can be expressed in terms of p=AE+iχH and m=BE−iχH, where $A=6\pi \varepsilon /\beta^{3}a_{n},B=6\pi \mu /\beta^{3}b_{n}$, and $\chi =6\pi n/c\beta^{3}c_{n}$ separately stand for the complex electric, complex magnetic, and electromagnetic dipole polarizabilities (which we call the chirality factor) of the spherical chiral particle with the scattered coefficients $a_{n},b_{n}$, and $c_{n}$ (note that the polarizabilities of the dipolar chiral particle are from ref. [54]). For an electromagnetic mixing through chirality factor χ, the chiral dipoles would experience the achiral and chiral forces in the gradient field, in which such a chiral force $\langle F_{\chi }\rangle$ could be decomposed into a reaction part and a dissipation part, which depend on the real and imaginary part of the complex chirality factor χ, respectively.
(12)$$\langle F_{\chi }\rangle =\langle F_{r}\rangle +\langle F_{d}\rangle ,$$
in which the component of reactive chiral force is directly related to the chiral density of the harmonic field given in Equation (Equation 9). One can thus write this as [55]:(13)$$\langle F_{r}\rangle =Re\left[c\chi \right]\left(c/\omega \right)\left[\nabla g\right],$$
and the component of dissipative chiral force is associated with the chiral flow given in Equation (Equation 8) and the time-averaged Poynting vector P, which can be written as [55]:(14)$$\langle F_{d}\rangle =Im\left[c\chi \right]\left(c/\omega \right)\left[\Phi -\frac{\mu }{4}\nabla \times P\right],$$
with $P=\frac{\mu }{2}Re\left(E\times H^{*}\right)$. The direct correlation between dissipative force and chirality flow suggests that a high chirality flow will enhance the interaction between light and chiral molecules.

From the expression of reactive chiral force given in Equation (Equation 13), it is clear that this conservative force corresponds to the gradient force $\langle F_{grad}\rangle$ induced by the interaction between particles with the chiral field, whose reaction component is proportional to the in-phase component of the chirality factor χ [55]. Consequently, the realization of this chiral force necessitates a strong optical chiral gradient [56]. In contrast, the dissipative achiral force described in Equation (Equation 14) is not conservative, and its dissipative nature results from the out-of-phase component of the chirality factors χ [55]. Actually, Equation (Equation 14) can be interpreted as analogous spin–orbit separation, wherein the first term corresponds to spin density force $\langle F_{spin}\rangle$ related to the SAM densities and chirality factor χ. The second term is the curl of the Poynting vector, which represents the vortex force $\langle F_{vor}\rangle$ determined by the energy flow vortex and chirality factor χ. Notably, the vortex force $\langle F_{vor}\rangle$ and spin density force $\langle F_{spin}\rangle$ would vanish if chirality (χ=0) were absent [51].

This paper explores the influence of a complex chirality factor χ on the optical forces exerted on a Rayleigh particle by a chiral focused field, determining the reactive chiral force (Equation (Equation 13)) and dissipative chiral force (Equation (Equation 14)) induced on the chiral Rayleigh particle immersed in a chiral focused field characterized by Equation (Equation 6).

### Numerical Results

Based on Equation (Equation 6), Figure 2 depicts the chiral focused field of the incident CV vortex beam through a 4π configuration. In what follows, the incident wavelength is λ=532 nm, the refractive index of the high NA lenses is n=1.518, the input power is P=100 mW, the numerical aperture is NA=1.26, the order of Bessel function s=1, and the phase delay is Δφ=π/2 unless otherwise specified. The lateral component $|E_{r}{|}^{2}$, longitudinal component $|E_{z}{|}^{2}$, and the total intensity ${\left|E\right|}^{2}$ of the electric-field distribution in the focal and axial planes are depicted in Figure 2a–c and Figure 2e–g, respectively. Specifically, the lateral focused field $|E_{r}{|}^{2}$ component in Figure 2a is obtained from the focusing of the azimuthally polarized part, whereas the axial component $|E_{z}{|}^{2}$ in Figure 2b is derived from the focusing of the radially polarized part. The total focused field ${\left|E\right|}^{2}$ illustrated in Figure 2c is, of course, the superposition of the transverse and axial components, which exhibits a doughnut-shaped intensity profile. Opposite sides of the focused field are out of phase, resulting in the on-axis energy null distribution as the beam propagates, as seen in Figure 2e–g. The phase distributions of the left and right total focused fields with the topological charge m=±1 are shown in Figure 2d and Figure 2h, respectively. The periodic phase distribution along the azimuth direction reveals that the focused field carries orbital angular momentum (OAM). Along the radial axis, the iso-phase presents a scroll-like shape in the inner core and a π/2-phase leap in the outside ring.

Prior to depicting the optical chiral effect, it is necessary to consider the properties of the optical chirality density of the focused field. The optical chirality density distributions of the focused field with topological charge m=±2 in the focal and axial planes are exhibited in Figure 3a,b and Figure 3d,e, respectively. It is found that the vortex beam with spatial polarization distribution provides a more flexible chirality density for the chiral focused field. Taking the chiral focused field with m=2 as an illustration, the full widths at half maximum (FWHM) in the lateral and longitudinal directions are computed to be 0.43λ and 0.65λ, respectively, and its size is roughly $\lambda^{3}/28$, which is significantly smaller than the diffraction limit of $\lambda^{3}/8$. By suppressing the transverse electric field component and boosting the longitudinal electric field component in the focal domain, it is envisaged that a small-scale focusing that exceeds the diffraction limit can be achieved [48]. Figure 3c illustrates the line scans along the radial direction of the chirality density of the focused field with various topological charges *m* in the focal planes. It is observed that, when the magnitude of topological charge *m* remains constant, the chiral optical vortex switches from clockwise to counterclockwise, and the central annular spot enlarges. Previous analysis indicates that the chiral focused field’s properties depend on the coherent superposition of the incident light field’s radial and azimuthal polarization components after it has passed through a high NA lens. Aiming at this factor, we depicted the radial optical chirality density as a function of the interception ratio $\beta_{0}$ of the high NA lens for the chiral focused field; these findings are illustrated in Figure 3f. It can be seen that the optical focal field has the maximum chiral density when the interception ratio $\beta_{0}$ falls between 1.0 and 1.5. Consequently, appropriately altering the interception ratio $\beta_{0}$ could enable the focused region to receive higher frequency components, thereby facilitating local chiral enhancement.

To further elucidate the chiral enhancement effect of the focused chiral field, Figure 4 depicts the optical superchirality distributions of the focused field in the focal and axial planes, for which the representative topological charge modes m=±2 and 0 were chosen. The focused field exhibits a significant chiral enhancement effect near the focal point, as seen in Figure 4a–c. When m=−2, the superchirality factor $g/g_{CL}$, compared to the CPL, can reach 2.5, and when m=2, the $g/g_{CL}$ can be raised to 3.9 or even higher. For an accurate evaluation of the optical superchirality enhancement, the magnetic dipole transition needs to be considered in the model [17,57]. Nevertheless, when m=0, the range of the superchirality factor is always maintained at $-1<g/g_{CL}<1$. Furthermore, as illustrated in Figure 4d–f, the localized superchiral field is constrained by a certain wavelength region. In the case of m=−2, the longitudinal distribution range of the superchiral field is −1.2λ<z<1.2λ, while in the situation of m=2, the focal depth of the superchiral field could be greater than 4λ, and the longitudinal region expands as the topological charge number *m* increases.

Next, we examine how the interaction between the chiral focused field and the manipulated Rayleigh particles affects the way they move. To minimize the competing achiral gradient force components in the focal plane, the electric and magnetic dipoles in the radial direction are chosen. Based on the optical force derived from the dipole approximation, we may precisely evaluate the force and analyze how it affects the chirality of the particles in the chiral focused field. Here, it is assumed that a chiral Rayleigh particle with radius r=10 nm and refractive index $n_{1}=1.59+0.005i,(\varepsilon_{a}=\varepsilon_{0}n_{2},\varepsilon_{0}=1.5$), suspended in a water chamber with refractive index n1=1.33, is illuminated by the chiral CV beams with the wavelength λ=532 nm [50,51,58]. (Note that the thermal mechanism is a factor that may destabilize the optical tweezers. It has been reported that the temperature of the water chamber near the surface of a Rayleigh particle needs to be kept at about 300 K [59].) At this time, the particle will be subjected to a force along the direction of the negative gradient of the electric field, which is the so-called “gradient force”. To capture particles stably, a condition that must be met is that the gradient force $\langle F_{grad}\rangle$ is greater than the scattering force $\langle F_{scat}\rangle$. Depicted in Figure 5a,b and Figure 5d,e are the lateral optical force experienced by the particle in the focal and axial planes illuminated by the highly focused chiral field, respectively. The background represents the total energy distribution of the chiral focused field, while the length and direction of the blue arrows indicate the magnitude and direction of the lateral optical force. In the field focusing region, the motion of particles is affected by the induced average optical force, which is caused by the transfer of linear momentum between light and matter. When light carries a vortex wavefront, particles experience a longitudinal orbital moment, causing them to rotate around the optical axis [60,61]. The orbital motion rotates counterclockwise on the inner ring with maximal intensity, clockwise on the outer ring with minimal intensity, and then vanishes from the central region where the intensity is slightly less than the maximum. For the case of χ=0.5−0.5i and m=−2 or 2, a stable transverse trap exists at the off-focus equilibrium point, as shown in Figure 5a or Figure 5b. Except for the radial and azimuthal optical forces near the maximum intensity, the particle experiences strong axial optical forces, as depicted in Figure 5d,e. The sign of topological charge alters the direction of the *z* component of the photon’s orbital angular momentum. When *m* is positive, the particles orbit the optical axis counterclockwise, whereas when *m* is negative, the particle rotates in the opposite direction [60]. Further, the dependence of the lateral optical force on the complex chirality factor χ is considered. Assuming that the Im[χ] is fixed at −0.5 and Re[χ] increases from −0.5 to 0.3, the line scans of optical force with various Re[χ] along the radial direction are illustrated in Figure 5c. Owing to the chiral focused field generated by two counter-propagating CV vortex light fields, the axial scattering force at the focal plane is counteracted, resulting in the formation of an equilibrium point along the axial direction at z=0 [50,58]. Meanwhile, asymmetric potential depth is formed in the lateral and longitudinal directions between the equilibrium points, which grows as the value of chirality factor Re[χ] rises. The stability of optical trapping at this point can be evaluated by a numerical calculation of the potential depth as U=−∫Fds [51]. The sectional view of the optical chirality density as functions of the chirality factor χ is depicted in Figure 5f, in which the line scan inserted in the figure compares the magnitude of optical force along the radial direction when the parameters Re[χ]=0.2,0.4,0.8 and Im[χ]=0.5. When the in-phase component of χ is $Re[\chi_{z}]=0.4$, there is a maximum optical force [50].

Previous analysis indicated that the reactive chiral force $\langle F_{r}\rangle$ derived from Equation (Equation 13) is dependent on the sign and magnitude of Re[χ] and on a large optical chiral gradient. The component of this force is proportional to the in-phase component of χ, which is associated with optical rotation in chiral systems. Figure 6 examines the effects of the sign and magnitude of the in-phase component of χ on the optical force exerted on the Rayleigh particle. The reactive chiral forces exerted on the particle with chirality factors χ=−0.5+0.1i and χ=0.5+0.1i in the focal and axial planes are illustrated in Figure 6a,b and Figure 6d,e, respectively. Provided that a chiral dipole has a chirality factor of the non-zero imaginary part, the transfer of momentum from the light to the particles when illuminated by the chiral light would induce a chiral torque $\langle \Gamma_{r}\rangle =\frac{{\left|E\right|}^{2}}{c}Im\left[c\chi \right]P$. Figure 6c depicts the influence of the gradient force and scattering force experienced by particles on chirality factors, in which the dotted red (χ=−0.5+0.1i) and blue (χ=0.5+0.1i) lines indicate the gradient force, and the dotted black line indicates the scattering force with corresponding parameters. It can be found that the axial scattering force is canceled and an equilibrium point is formed along the longitudinal axis at z=0. Moreover, the OAM mode carried by the chiral focused field enhances the axial and radial gradient forces, and the maximum displacement of a single potential well is estimated to be λ [51]. The chiral focused field exerts diverse transverse optical forces on dipolar chiral particles with various chiral factors. The successive optical force induces the particle to rotate around the optical axis and to follow different trajectories, enabling efficient optical sorting of chiral particles with complex motion patterns such as the spiral pattern illustrated in Figure 6f.

For pursuing the physical origin of dissipative optical force on chiral particles, Figure 7 depicts the dissipative chiral force experienced by the particle, with an emphasis on the influences of chirality factor χ and topological charge *m* on the dissipative chiral force. The chiral focal field carrying OAM has a π/2 phase difference between radial and azimuthal field components, which induces the chiral flow to bend in a spiral orbit around the inner ring, thus driving the particles to experience a longitudinal optical torque and rotate around the optical axis. Actually, the chirality flow can be regarded as a similar spin–orbit separation $\Phi =\Phi_{spin}+\Phi_{orbit}$. Provided that the dipole is chiral and the chirality flow is non-zero, the torque would be exerted independently of the chiral nature of the focused field. Comparing Figure 7a,c with Figure 7d,f, it can be found that, when the topological charge changes from m=−2 to m=2, the direction of chirality flow of the dissipative chiral forces reverses and alternates between positive and negative. This signifies that the sign of the topological charge determines both the orientations of the SAM and OAM densities of the chiral focused field as well as the spinning direction of the particle. Nevertheless, when the topological charge m=0 and the chirality flux Φ=0, the total field loses its chiral property, and the chiral dipole illuminated by this field would not experience the chiral optical moment. Additionally, the total Poynting vector will dominate the dissipative chiral force and induce the particles to rotate around the optical axis, as illustrated in Figure 7b,d. This phenomenon proves that the orbital motion of chiral particles induced by lateral dissipative chiral forces derives from the transformation of the localized OAM of the chiral field, which is also affected by the out-of-phase component of the chirality factor χ of particles. The dissipative component of the chiral force is not conservative, which is determined by the chirality flow Φ, the electromagnetic orbital parts of the time-averaged Poynting vector P, and the out-of-phase component of χ that is associated with circular dichroism [55].

## 3. Conclusions

In conclusion, a chiral focused field could be generated by modulating the spiral wavefront of two counter-propagating CV vortex beams in a 4π microscope system, of which superchiral enhancement $g/g_{CL}$ is 3.9-fold greater than that of a CPL field. It has been found that the optical chirality of the focused field depends not only on the sign and size of the photon OAM carried by the incident light but also on the interception ratio of the high NA lens. When the interception ratio is within the range of $1<\beta_{0}<1.5$, there is an apparent superchiral enhancement effect near the focal point, where the FWHM in the lateral and longitudinal directions are calculated to be 0.43λ and 0.65λ, respectively, and its size is about $\lambda^{3}/28$. The region with enhanced superchirality extends as the increase of topological charge *m*. Additionally, the chiral mechanical effect is characterized by the behavior of optical chirality density *g* and flow Φ. In the dipolar framework, the reactive and dissipative components of the chiral force would be separately associated with the motion of optical rotation (Re[χ]) and circular dichroism (Im[χ]). The chiral focused field can induce various lateral reactive and dissipative chiral forces on the dipolar chiral particles with different chirality, which are related to the in-phase component (associated with *g*) and out-of-phase component (associated with Φ) of the chirality factor χ, respectively. Results obtained would provide a clear understanding of the interaction between the light and chiral particles and further enable effective optical sorting and accurate capture of particles with complex motion trajectories.

## Figures and Tables

**Figure 1 nanomaterials-13-02251-f001:**
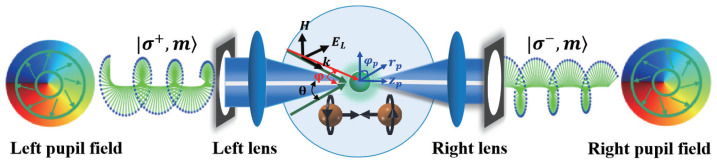
Schematic of the microparticle manipulation using the 4π focusing system.

**Figure 2 nanomaterials-13-02251-f002:**
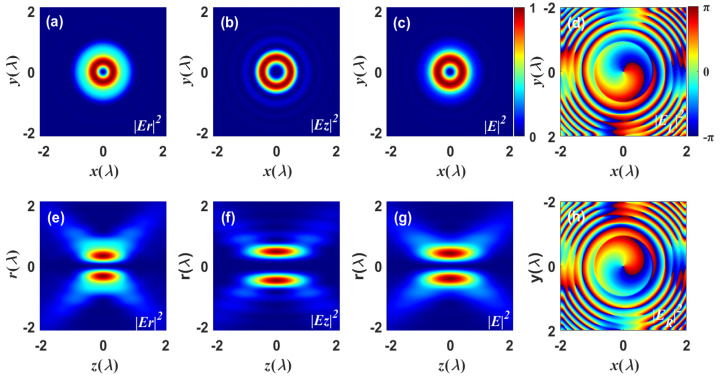
Snapshots illustrate the focused field of the vector vortex beam. (**a**–**c**) The electric-field intensity distributions in the focal planes; (**e**–**g**) the electric-field intensity distributions in the axial planes; (**d**,**h**) the phase distributions of the left and right total fields. The parameters specified are m=1 and $\beta_{0}=1$.

**Figure 3 nanomaterials-13-02251-f003:**
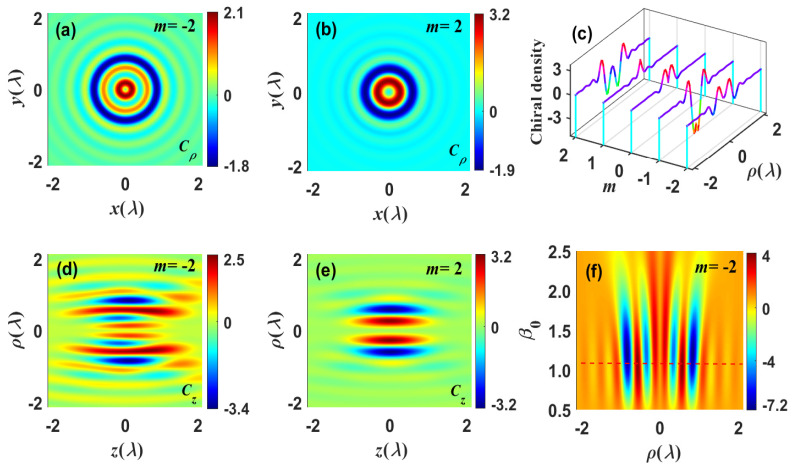
Snapshots illustrate the optical chirality density of the focused field. (**a**,**b**) and (**d**,**e**) represent the optical chirality density distributions of the focused field with different topological charges *m* in the focal and axial planes, respectively. (**c**) Line scans along the *x* axes of the longitudinal optical chirality density of the focused field with different topological charges *m* in the focal planes. (**f**) The radial optical chirality density as functions of $\beta_{0}$.

**Figure 4 nanomaterials-13-02251-f004:**
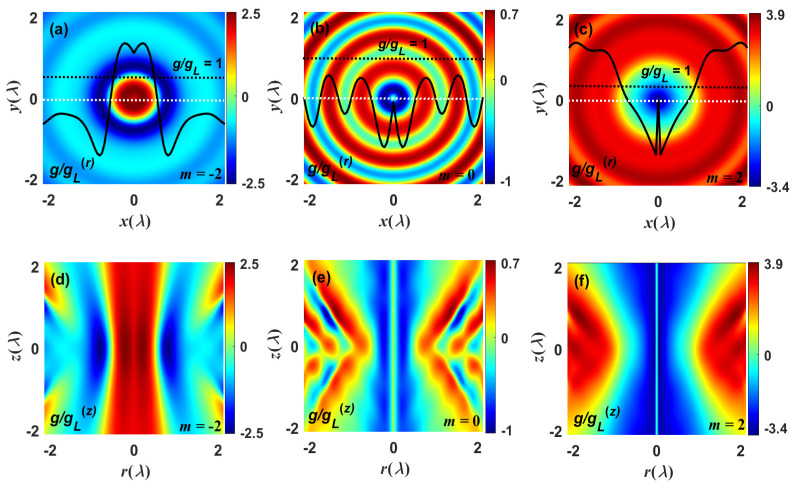
Snapshots describe the superchirality density of the focused field. (**a**–**c**) and (**d**–**f**) represent the optical superchirality density distributions of the focused field with different topological charges *m* in the focal and axial planes, respectively. The insets shown in (**a**–**c**) are line scans of the superchirality density along the radial direction, and the black dotted line indicates the position where the value of the superchirality density $g/g_{CL}$ is equal to 1.

**Figure 5 nanomaterials-13-02251-f005:**
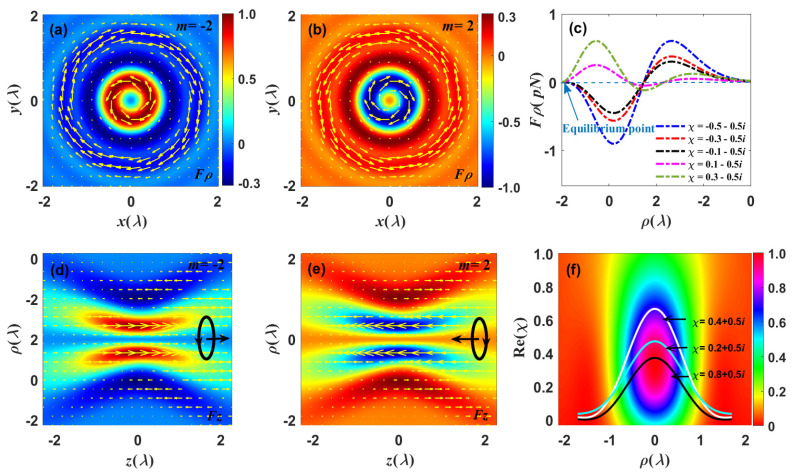
Sectional views of the lateral optical force experienced by the particle during chiral focused field illumination. (**a**,**b**) and (**d**,**e**) represent the optical force distributions of the focused field with different topological charges *m* in the focal and axial planes, respectively. (**c**) Line scans along the radial direction of the optical force with different chirality factor χ in the focal and axial planes. (**f**) Sectional views of the radial optical chirality density as functions of the chirality factor χ.

**Figure 6 nanomaterials-13-02251-f006:**
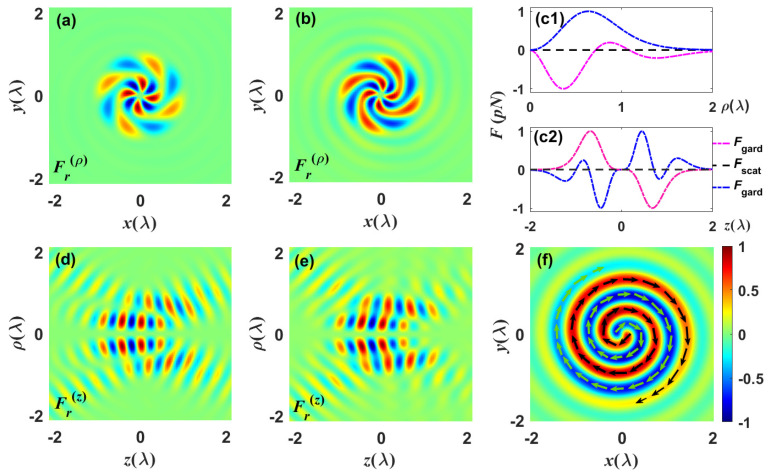
Snapshots describe the reactive chiral force experienced by the particle with various χ chirality factors. The reactive chiral force exerted on a particle with chirality factors χ=−0.5+0.1i (**a**) and χ=0.5+0.1i (**b**) in the focal plane, respectively. Optical forces generated by a chiral field acting on Rayleigh particles along (**c1**) radial (**c2**) longitudinal axes, where the dotted red line Re[χ]=−0.5 and the dotted blue line Re[χ]=0.5. (**d**,**e**) Sectional views of the longitudinal reactive chiral force corresponding to (**a**,**b**) along the axial planes. (**f**) Trajectory of chiral particles along the spiral path on the focal plane; the orientation of its tension is indicated by an arrow.

**Figure 7 nanomaterials-13-02251-f007:**
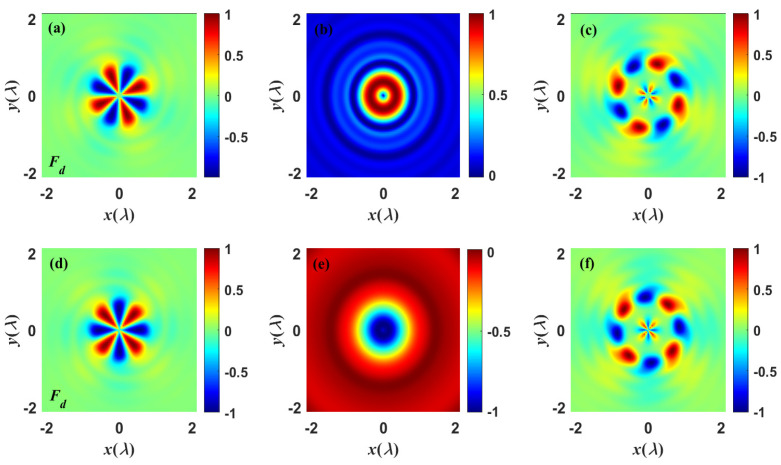
Snapshots describe the dissipative chiral force experienced by the particle with various chirality factors χ and topological charge *m*. Upper row: (**a**–**c**) χ=0.5−0.5i, (**a**) m=−2, (**b**) m=0, (**c**) m=2. Lower row: (**d**–**f**) χ=0.5+0.5i, (**d**) m=−2, (**e**) m=0, (**f**) m=2.

## Data Availability

No data were used for the research described in the article.

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
