# Peer review of "Chiral Mechanical Effect of the Tightly Focused Chiral Vector Vortex Fields Interacting with Particles"

_nanomaterials, 2023, doi:10.3390/nano13152251_

Round 1

Reviewer 1 Report

The manuscript describes a numerical study of the optical force exerted by two counterpropagating focused vortex beams on a chiral particle in the Rayleigh approximation. The authors demonstrated that the experimental configuration they describe can be used for chiral particle separation and identified a superchiral regime at which the superchiral enhancement of the force can be observed. The paper can be of interest to a broader photonics community and can be published after minor revisions. Specifically:

1. The first reference [1] may not be the best to define chirality of an object. I would also re-phrase the definition of chirality as "a structure is not superimposable with its mirror image".

2. Line 16. Chirality results from the electric and magnetic vector fields... If the authors mean chiraly of light, as defined in Ref. [16], it should be specified.

3. Line 41. What is meant by "the size of the light"? This definition is not physical. If the authors mean the wavelength of light it should be said so.

4. Lines 43 through 46. This whole sentence is not clear and should be re-phrased for clarity.

5. Formula (7) must be corrected according to Ref. [16]. The permeability of vacuum is supposed to be in the denominator.

6. Formula (9) and the sentence right above it. Eq. (7) cannot be simplified as Eq. (9) because C is defined as optical chirality and g is defined as chirality density. I suggest the authors introduce chirality density in Eq. 7 instead of optical chirality.

7. Line 133. Light exerts instantaneous force and momentum, not moment.

8. Line 139. Why is the electric dipole moment p defined in two different ways? It looks like it should be p and m, electric and magnetic dipoles.

Extensive editing of English language is required.

Author Response

Response to the referees’ comments

Reviewer: 1

Comments to the Author

The manuscript describes a numerical study of the optical force exerted by two counterpropagating focused vortex beams on a chiral particle in the Rayleigh approximation. The authors demonstrated that the experimental configuration they describe can be used for chiral particle separation and identified a superchiral regime at which the superchiral enhancement of the force can be observed. The paper can be of interest to a broader photonics community and can be published after minor revisions. Specifically:

  1. The first reference [1] may not be the best to define chirality of an object. I would also re-phrase the definition of chirality as "a structure is not superimposable with its mirror image".

Response: Thank you for your professional comment. The content “Chirality is defined as a structure that is not superimposable with its mirror image” has been modified in the revised manuscript. (Please see the revised manuscript in the line 14 of the page 1).

  1. Line 16. Chirality results from the electric and magnetic vector fields... If the authors mean chiraly of light, as defined in Ref. [16], it should be specified.

Response: We are very sorry for our negligence.

Upon careful consideration of your suggestion and after consulting Ref. [16], we agree that the term "chirality" used in our manuscript pertains specifically to the chirality of light, as defined in Ref. [16].

To ensure clarity, we have made necessary revisions in the manuscript to specify that we are referring to the chirality of light. The content “Chirality of light” have been added. (Please see the revised manuscript in the line 16 of the page 1).

  1. Line 41. What is meant by "the size of the light"? This definition is not physical. If the authors mean the wavelength of light it should be said so.

Response: We are immensely grateful for your attention to detail and for pointing out this crucial aspect that required clarification.

The content “the size of the light” has been modified as “the dimension of vortex beams”. (Please see the revised manuscript in the line 40 of the page 2).

  1. Lines 43 through 46. This whole sentence is not clear and should be re-phrased for clarity.

Response: Based on your suggestion, we have rephrased the sentence for better clarity:

“The OAM states induced by chirality would promote the superchiral response of photon spin states to excited molecules, and realize the exact capture of nanoparticles with complex trajectories” has been modified. (Please see the revised manuscript in the line 42-45 of the page 2).

We believe this revision provides a more straightforward and comprehensible explanation of the intended concept. 

  1. Formula (7) must be corrected according to Ref. [16]. The permeability of vacuum is supposed to be in the denominator.

Response: Regarding your concern about Formula (7) in our manuscript, you mentioned that it should be corrected according to Ref. [16], with the permeability of vacuum in the denominator. After carefully reevaluating both our formula and the reference you provided, we believe that Formula (7) is correct as stated in our manuscript according to the following Refs. [1-4]. We have thoroughly cross-referenced it with the relevant literature, including Ref. [16], and found no discrepancy in its current form.

[1] Li, M.; Yan, S.; Zhang, Y.; Yao, B. Generation of controllable chiral optical fields by vector beams. Nanoscale 2020, 12, 15453–15459.

[2] Bohren, C. F.; Huffman, D. R. Absorption and Scattering of Light by Small Particles. John Wiley & Sons, New York, 1983.

[3] Lindell, I. V.; Sihvola, A. H.; Tretyakov, S. A.; Viitanen, A. J. Electromagnetic Waves in Chiral and Bi-Isotropic Media, Artech House, Boston, 1994.

[4] Lakhtakia, A.; Varadan, V. K.; Varadan, V. V. Time-Harmonic Electromagnetic Fields in Chiral Media, Springer-Verlag, New York, 1989.

  1. Formula (9) and the sentence right above it. Eq. (7) cannot be simplified as Eq. (9) because C is defined as optical chirality and g is defined as chirality density. I suggest the authors introduce chirality density in Eq. 7 instead of optical chirality.

Response: Thank you for your constructive comments to our work!

The context “The eq.(7) could be simplified as ” has been deleted. And the formula of the optical chirality density has been modified as . (Please see the revised manuscript in the line 122 of the page 4).

  1. Line 133. Light exerts instantaneous force and momentum, not moment.

Response: Thank you for this professional comment.

The content “moment” has been modified as “momentum”. (Please see the revised manuscript in the line 131 of the page 4).

  1. Line 139. Why is the electric dipole moment p defined in two different ways? It looks like it should be p and m, electric and magnetic dipoles.

Response: We are very sorry for our negligence.

Here it should be “”, and we mistakenly wrote “”. (Please see the revised manuscript in the line 135 of the page 4).

We thank you for your thorough review and your commitment to enhancing the quality of our manuscript.

Reviewer 2 Report

The following sentences and equations should be corrected

Line # 79 check for subscript for Ei_theta instead of radial component

Line # 90 According to Richards wolf’s vector diffraction theory -> Richards–Wolf (B. Richards and E. Wolf “Electromagnetic diffraction in optical systems II. Structure of the image field in an aplanatic system,” Proc. Roy. Soc. A-Math Phy 253, 358–379 (1959). https://royalsocietypublishing.org/doi/abs/10.1098/rspa.1959.0200)

Line # 101 According to the vector Debye diffraction theory -> reference needed

Line # 103 Eq.(5) parentheses l(theta)

Line # 110  Note that the chiral focused field can only be realized when the component of the electric field is parallel or opposite to the magnetic field. -> What about circular polarized light in which the electric field and magnetic fields are orthogonal? Is the circular polarized light nonchiral?

Line 119 Eq.(9) contains g instead of C, hence it cannot be a simplification of Eq.(7)

Line # 126 For a minor chiral Rayleigh particle, such as a dielectric and magnetic spherical particle suspended in the space of the chiral focused field, whose optical properties could be described by the polarizability –> such particle is not chiral 

Line # 135 the force exerted on the inductive (electric(?))  dipole, the force exerted by(?) the inductive magnetic dipole, and the force generated by the interaction between (?) the two dipoles

Author Response

Reviewer: 2

Comments to the Author

The following sentences and equations should be corrected

  1. Line # 79 check for subscript for Ei_theta instead of radial component

Response: We are very sorry for our negligence.

In our revised manuscript, the context “” have been modified as . (Please see the revised manuscript in the line 78 of the page 2).

  1. Line # 90 According to Richards wolf’s vector diffraction theory-> Richards–Wolf (B. Richards and E. Wolf “Electromagnetic diffraction in optical systems II. Structure of the image field in an aplanatic system,” Proc. Roy. Soc. A-Math Phy 253, 358 – 379 (1959).  https://royalsocietypublishing.org/doi /abs/10.1098/rspa.1959.0200)

Response: We truly value your attention to detail and your commitment to ensuring the accuracy and validity of our research.

Regarding your comment on Line 90, where we mentioned the Richards wolf’s vector diffraction theory, we apologize for the oversight in not providing a specific reference to support this statement.

Ref. “[46] Richards, B.; Wolf, E. Electromagnetic diffraction in optical systems II. Structure of the image field in an aplanatic system. Proceedings of the Royal Society of London 1959, 253, 358-379. ” has been cited to strengthen the credibility of our work.

  1. Line # 101 According to the vector Debye diffraction theory -> reference needed

Response: We sincerely thank you for bringing this to our attention, as it has significantly improved the clarity and accuracy of our manuscript.

After revisiting our manuscript, we have identified the appropriate reference to cite for the vector Debye diffraction theory.

Ref. “[46] Richards, B.; Wolf, E. Electromagnetic diffraction in optical systems II. Structure of the image field in an aplanatic system. Proceedings of the Royal Society of London 1959, 253, 358-379. ”

And “[47] Novotny, L.; Hecht, B. Principles of Nano Optics. Cambridge University Press, New York, 2006.” has been cited to strengthen the credibility of our work.

  1. Line # 103 Eq.(5) parentheses l(theta)

Response: We are very sorry for our negligence.

In our revised manuscript, the context “” have been modified. (Please see the revised manuscript in the line 102 of the page 3).

  1. Line # 110  Note that the chiral focused field can only be realized when the component of the electric field is parallel or opposite to the magnetic field. -> What about circular polarized light in which the electric field and magnetic fields are orthogonal? Is the circular polarized light nonchiral?

Response: Thank you for this professional comment.

You have raised an important point, and we apologize for any oversight in our original explanation. Circular polarized light indeed is chiral light.

The inappropriate expressio “Note that the chiral focused field can only be realized when the component of the electric field is parallel or opposite to the magnetic field. ” has been removed. (Please see the revised manuscript in the line 110 of the page 3).

  1. Line 119 Eq.(9) contains g instead of C, hence it cannot be a simplification of Eq.(7)

Response: Thank you for your constructive comments to our work!

The context “The eq.(7) could be simplified as ” has been deleted. And the formula of the optical chirality density has been modified as . (Please see the revised manuscript in the line 122 of the page 4).

  1. Line # 126 For a minor chiral Rayleigh particle, such as a dielectric and magnetic spherical particle suspended in the space of the chiral focused field, whose optical properties could be described by the polarizability –> such particle is not chiral 

Response: We would like to express our heartfelt gratitude for your thorough review!

You are absolutely correct in noting that a minor chiral Rayleigh particle, such as a dielectric and magnetic spherical particle, is not intrinsically chiral. We apologize for any confusion that our original statement may have caused.

The context “For a minor chiral Rayleigh particle, such as a spherical dipolar particle suspended in the space of the chiral focused field” has been removed. (Please see the revised manuscript in the line 124 of the page 4).

  1. Line # 135 the force exerted on the inductive (electric(?))  dipole, the force exerted by(?) the inductive magnetic dipole, and the force generated by the interaction between (?) the two dipoles

Response: We would like to express our gratitude for your meticulous review of our manuscript.

Regarding your comment on Line #135, where you pointed out that the terminology used in that section is incorrect, and we apologize for any confusion it may have caused. To address this, we have carefully reevaluated the mentioned section and made the necessary revisions to provide a more accurate and coherent explanation.

The context “ the superposition of three forces: the force exerted on the inductive dipole, the force exerted by the inductive magnetic dipole, and the force generated by the interaction between the two dipoles ” has been removed. (Please see the revised manuscript in the line 133 of the page 4).

And the context “Thus, the vector expression of the optical force acting on the particles can be expressed as” has been modified (Please see the revised manuscript in the line 132 of the page 4).

Reviewer 3 Report

In this paper the authors produced a chiral focused field by modulating the spiral wavefront of two counter-propagating CV vortex beams in a microscope system. They studied the optical chirality of the focused field showing that this depends on the sign and size of the photon OAM carried by the incident light and also on the interception ratio of the high NA lens. When the interception ratio is within the range of 1 < b0 < 1.5, there is an apparent superchiral enhancement effect near the focal point. The region with enhanced superchirality extends as the increase of topological charge m. Additionally, the chiral mechanical effect is characterized by the behavior of optical chirality density g and flow F. In the dipolar framework, the reactive and dissipative components of the chiral force would be separately associated with the motion of optical rotation (Re[c]) and circular dichroism (Im[c]). The chiral focused field can induce various lateral reactive and dissipative chiral forces on the dipolar chiral particles with different chirality, which are related to the in-of-phase component(associated with g) and out-of-phase component (associated with F) of the chirality factor c, respectively.

the results here reported provide an additional understanding of the interaction between the light and chiral particles and further enable effective optical sorting and accurate capture of particles with complex motion trajectories.

The manuscript is very well written and it sounds interesting. 

According to my expertiese on the topic I think that it could be maybe useful to mention some important papers where nanoparticles interacting with OAM carrying beam were studied. See for example:

Nano Lett. 17, 7920 (2017).

Nanophotonics 8, 1227 (2019).

Opt. Lett. 45, 823-826 (2020).

Nat. Nanotechnol. 9, 295 (2014).

Author Response

Reviewer: 3

Comments to the Author

In this paper the authors produced a chiral focused field by modulating the spiral wavefront of two counter-propagating CV vortex beams in a microscope system. They studied the optical chirality of the focused field showing that this depends on the sign and size of the photon OAM carried by the incident light and also on the interception ratio of the high NA lens. When the interception ratio is within the range of 1 < b0 < 1.5, there is an apparent superchiral enhancement effect near the focal point. The region with enhanced superchirality extends as the increase of topological charge m. Additionally, the chiral mechanical effect is characterized by the behavior of optical chirality density g and flow F. In the dipolar framework, the reactive and dissipative components of the chiral force would be separately associated with the motion of optical rotation (Re[c]) and circular dichroism (Im[c]). The chiral focused field can induce various lateral reactive and dissipative chiral forces on the dipolar chiral particles with different chirality, which are related to the in-of-phase component(associated with g) and out-of-phase component (associated with F) of the chirality factor c, respectively.

the results here reported provide an additional understanding of the interaction between the light and chiral particles and further enable effective optical sorting and accurate capture of particles with complex motion trajectories.

The manuscript is very well written and it sounds interesting.

According to my expertiese on the topic I think that it could be maybe useful to mention some important papers where nanoparticles interacting with OAM carrying beam were studied. See for example:

  1. Nano Lett. 17, 7920 (2017).
  2. Nanophotonics 8, 1227 (2019).
  3. Lett. 45, 823-826 (2020).
  4. Nanotechnol. 9, 295 (2014).

Response: We sincerely appreciate your expertise and insightful review of our manuscript. Your suggestion to mention important papers that study nanoparticles interacting with OAM carrying beams is highly valuable, and we thank you for bringing this to our attention.

In response to your comment, we have carefully considered your recommendation and agree that referencing relevant and influential papers on this topic would enhance the value and relevance of our research. We will cite those references in our manuscript dedicated to citing these important works.

Ref. “[43] Huft, P. R.; Kolbow, J. D.; Thweatt, J. T.; Lindquist, N. C. Holographic plasmonic nanotweezers for dynamic trapping and manipulation. Nano Letters 2017, 12, 7920–7925.

[44] Liu, K.; Maccaferri, N.; Shen, Y.; Li, X.; Zaccaria, R.;, Zhang, X.; Gorodetski, Y.; Garoli, D. Particle 449 trapping and beaming using a 3D nanotip excited with a plasmonic vortex. Optics Letters 2020, 45, 823-826. 451

[45] Kotsifaki, D.; Chormaic, S. Plasmonic optical tweezers based on nanostructures: fundamentals, advances and prospects. Nanophotonics 2019 , 8, 1227-1245” has been cited 
